# Sources of noise during accumulation of evidence in unrestrained and voluntarily head-restrained rats

**Benjamin B Scott[1,2†], Christine M Constantinople[1,2†], Jeffrey C Erlich[3], David W Tank[1,2,4]\*, Carlos D Brody[1,2,5]\***

[1]Princeton Neuroscience Institute, Princeton University, Princeton, United States; [2]Department of Molecular Biology, Princeton University, Princeton, United States; [3]NYU-ECNU Institute of Brain and Cognitive Science, New York University Shanghai, Shanghai, China; [4]Bezos Center for Neural Circuit Dynamics, Princeton University, Princeton, United States; [5]Howard Hughes Medical Institute, Princeton University, Princeton, United States

**Abstract** Decision-making behavior is often characterized by substantial variability, but its source remains unclear. We developed a visual accumulation of evidence task designed to quantify sources of noise and to be performed during voluntary head restraint, enabling cellular resolution imaging in future studies. Rats accumulated discrete numbers of flashes presented to the left and right visual hemifields and indicated the side that had the greater number of flashes. Using a signal-detection theory-based model, we found that the standard deviation in their internal estimate of flash number scaled linearly with the number of flashes. This indicates a major source of noise that, surprisingly, is not consistent with the widely used 'drift-diffusion modeling' (DDM) approach but is instead closely related to proposed models of numerical cognition and counting. We speculate that this form of noise could be important in accumulation of evidence tasks generally.

**\*For correspondence:** dwtank@princeton.edu (DWT); brody@princeton.edu (CDB)

[†]These authors contributed equally to this work

**Competing interests:** The authors declare that no competing interests exist.

## Introduction

Subjects performing perceptual decision-making tasks display a large amount of trial-to-trial behavioral variability. Determining the sources of this variability could provide insight into the neural mechanisms of decision-making and produce more accurate predictions of behavior. It has been proposed that behavioral variability is caused in part by noise accruing during the process of evidence accumulation. This noise may have a variety of origins depending on the behavioral task. It can be inherent in the natural world, produced by the signal detection limits of sensory organs themselves (*Barlow and Levick, 1969*), or it may reflect the variability of neural responses in the brain at different stages of processing. Previous studies have attempted to trace the sources of noise using a combination of behavioral and neurophysiological approaches.

At the behavioral level, many models of the evidence accumulation process are known as 'drift-diffusion' models (DDMs), because a major component of the noise is modeled as diffusion noise in the accumulator, i.e., noise that is independent for each time point (*Bogacz et al., 2006*; *Smith and Ratcliff, 2004*, but see *Zariwala et al., 2013*). A recent study by Brunton and colleagues (*Brunton et al., 2013*) developed an accumulation of evidence task, which they modeled with a DDM-like process, to isolate different sources of noise. In their task, evidence was delivered in randomly timed but precisely known pulses: two randomly generated streams of discrete auditory clicks were presented from left and right speakers, and subjects (rats and humans) were trained to indicate the side with the greater number of clicks. The precise timing of the stimuli combined with large

**eLife digest** Perceptual decision-making, i.e. making choices based on observed evidence, is rarely perfect. Humans and other animals tend to respond correctly on some trials and incorrectly on others. For over a century, this variability has been used to study the basis of decision-making. Most behavioral models assume that random fluctuations or 'noise' in the decision-making process is the primary source of variability and errors. However, the nature of this noise is unclear and the subject of intense scrutiny.

To investigate the sources of the behavioral variability during decision-making, Scott, Constantinople et al. trained rats to perform a visual 'accumulation of evidence' task. The animals counted flashes of light that appeared on either their left or their right. Up to 15 flashes occurred on each side, in a random order, and the rats then received a reward if they selected the side that the greatest number of flashes had occurred on. The rats chose correctly on many occasions but not on every single one.

Using a computer-controlled rat training facility or 'rat academy', Scott, Constantinople et al. collected hundreds of thousands of behavioral trials from over a dozen rats. This large dataset provided the statistical power necessary to test the assumptions of leading models of behavioral variability during decision-making, and revealed that noise grew more rapidly with the number of flashes than previously predicted. This finding explained patterns of behavior that previous models struggled with, most notably the fact that individuals make errors even on the easiest trials. The analysis also revealed that animals maintain two separate running totals – one of stimuli on the left and another of stimuli on the right – rather than a single tally of the difference between the two.

Scott, Constantinople et al. further demonstrated that rats could be trained to perform this task using a new system that enables functional brain imaging. The next step is to repeat these experiments while simultaneously recording brain activity to study the neural circuits that underlie decision-making and its variability.

numbers of behavioral trials enabled fitting of a detailed and statistically powerful behavioral model. The results suggested that in most subjects the diffusion noise in the accumulator was essentially zero (noiseless), and the behavioral variability was best explained by noise that was added with each pulse of evidence. However, it was assumed that each pulse introduced independent noise. This implied that although time-locked to the stimulus, the noise associated with the stimulus was diffusion-like, and the standard deviation of the total stimulus-induced noise scaled as the square root of the number of pulses. Overall, the model was qualitatively consistent with the widely-assumed diffusion-like noise in the evidence accumulation process. However, similar to other reports, diffusion-like noise alone was unable to fully predict behavioral variability, and specifically failed to account for errors on the easiest trials. Following common practice, *Brunton et al. (2013)* included an additional noise parameter, termed lapse rate, which described variability that did not depend on the stimulus or trial duration. The need for the lapse rate term suggests the existence of additional sources of noise that are not diffusion-like.

To better characterize sources of noise during perceptual decision-making, here we developed a visual analogue of the Brunton et al. accumulation of evidence task with two key features (*Erlich et al., 2013*, SFN, abstract). First, rats could perform the task during voluntary head-restraint (*Girman, 1980*; *1985*; *Kampff et al., 2010*, SFN, abstract), allowing for the potential of cellular resolution imaging (*Scott et al., 2013*) and perturbation in future studies (*Rickgauer et al., 2014*). Second, the sensory stimuli were designed to attempt to isolate internally generated noise by minimizing noise inherited from the stimulus. In our task, rats were presented with two series of brief (10 ms), pseudorandomly timed LED flashes to the left and right visual hemifields. Following the form of the *Brunton et al. (2013)* task, the side with the greater total number of flashes indicated the location of a water reward. However, whereas the acoustics of the behavior chamber could distort localization of auditory stimuli, here our visual stimuli were presented from well-separated LEDs positioned to the left and right visual hemifields, suggesting that the rats would have no difficulty

distinguishing the left versus right origin of each flash. Additionally, low numbers of flashes were presented in well-separated time bins.

Behavioral analyses indicated that rats solved the task by accumulating all available evidence, and that flash-associated noise was the predominant source of behavioral variability. We implemented a signal detection theory-based model to evaluate the assumption that noise in the accumulator value was diffusion-like, i.e. that noise was independent for each flash and the noise (standard deviation) of the accumulator value scaled as the square root of the number of flashes. The signal detection theory-based model was designed to find the scaling relationship between noise in the accumulator value and number of flashes that best described the data. Surprisingly, model fits revealed that the noise in the subjects' numerical estimates scaled linearly with the number of flashes presented, not as the square root. This relationship, called scalar variability, has previously been observed in tasks that require subjects to estimate the duration of a stimulus (*Gibbon, 1977*) or count the number of stimuli (*Gallistel and Gelman, 2000*). Moreover, a behavioral model implementing scalar variability predicted imperfect behavioral performance on the easiest trials, without the need for a non-zero 'lapse rate' parameter. Linear scaling of the standard deviation of the noise with the number of flashes reveals a source of noise that does not treat individual pulses independently. Therefore, the noise is not diffusion-like; unlike a lapse rate it depends on total number of evidence pulses; and it is introduced after single-flash sensory processing. We suggest that taking into account this form of noise will be a critical factor for understanding variability in decision-making behaviors.

An orthogonal approach for measuring noise in the accumulation process is to relate the variability of neuronal responses to behavior (*Shadlen et al., 1996*; *Cohen and Newsome, 2009*; *Mazurek et al., 2003*). Ideally, this approach would include recordings from multiple neurons across

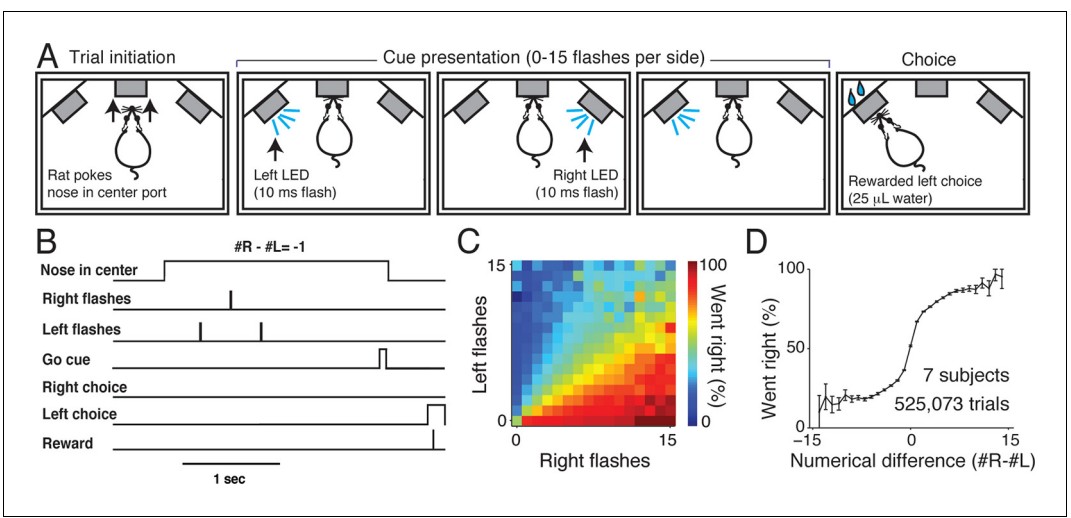

**Figure 1.** Visual accumulation of evidence task, unrestrained version. (A) Task schematic. A rat initiates a behavioral trial by inserting his nose into a center port along one wall of an operant training box (left panel). The rat is presented with a series of pseudo-randomly timed flashes from LEDs presented from the left and right side ports (middle panels). Following cue presentation and a variable delay period (see Materials and methods: Stimulus Generation), the subject makes a behavioral report by poking his nose to one of the side ports (right panel). Water reward (25 μL) was baited on the side that had the greater number of flashes. (B) Timing of task events in an example trial. Rat was presented with two left flashes and one right flash and correctly oriented to the left side poke after an auditory cue. (C) Behavioral performance of data pooled from all unrestrained rats across the set of all stimuli. Color indicates the percentage of trials in which the subjects chose the right sideport. (D) Psychophysical performance on the accumulation task pooled from all unrestrained rats.

The following figure supplements are available for figure 1:

**Figure supplement 1.** Stimuli and performance.

**Figure supplement 2.** Psychometric performance of individual rats on the visual accumulation of evidence task.

brain regions and precise neural circuit manipulations. To this end, we developed a version of our task that rats could perform during voluntary head restraint. Voluntary head restraint was developed as an alternative to forced head restraint in which the initiation and termination of the restraint period are under the control of the animal (*Girman, 1980*; *1985*; A.R. Kampff et al., 2010, SFN, abstract). Recently we reported that rats can be trained to perform voluntary head restraint in a high-throughput semi-automated behavioral facility and described a voluntary head restraint system that provides the stability needed for *in vivo* cellular resolution imaging (*Scott et al., 2013*). Here we show that performance of head restrained rats was essentially identical to the performance of rats trained on the unrestrained version of the task. These results demonstrate that rats can perform complex cognitive behaviors during voluntary head restraint and provide a platform for characterizing noise during decision-making across multiple brain regions.

## Results

In our rat visual accumulation of evidence task (*Figure 1A and B*), subjects initiate a trial by inserting their nose into the center port of a three-port operant conditioning chamber. Subjects must keep their nose in the center port (fixation) for 1–8 s while a series of brief (10 ms) flashes are presented by LEDs to the left and right visual hemifields. Between 0 and 15 flashes are presented independently to each hemifield with a randomized number and timing of flashes on each side (*Figure 1— figure supplement 1*). Trial durations and number of presented flashes are independently varied on a trial-by-trial basis. Following an auditory go cue that signals the end of fixation, subjects receive water reward (typically 25 µL) for orienting to a nose port on the side that had more flashes.

We trained seven rats to perform this task using an automated procedure in a high-throughput behavioral facility. Rats progressed through a series of stages in which they learned (1) to associate light with reward, (2) to maintain nose in center fixation for increasingly long durations and (3) to compare the number of flashes on each side to predict the rewarded location. The procedural code used for training can be downloaded from http://brodylab.org/code/flash-code. Rats progressed through the training stages in 3500 trials. Fully trained rats performed a combined 525,073 trials at 91% for the easiest trials and 70% correct overall. Moreover rats were sensitive to the difference in the number of flashes including differences of a single flash (*Figure 1C and D*; *Figure 1—figure supplement 2*).

### Errors in behavioral choice increase with number of flashes, not trial duration

Accumulation of evidence involves two processes: maintaining a memory of the evidence and adding new evidence to that memory. To assess whether noise (and thus behavioral variability) was more closely associated with the memory of the accumulator or with incoming sensory evidence, we initially fit the Brunton et al. model to our visual task data. This produced results consistent with those found for Brunton et al.'s auditory task, including near-zero estimates of accumulator memory noise (i.e., a predominant role for incoming sensory evidence noise), and long accumulation time constants (*Figure 2—figure supplement 1*). However, further analysis described below led us to question Brunton et al.'s assumption of independent noise across pulses of sensory evidence. We therefore took a model-free approach to estimate whether noise was more closely associated with the memory of the accumulator or with incoming sensory evidence. When trial duration and flash difference were held constant, errors increased with total number of flashes presented, suggesting that noise increased with each flash (*Figure 2A*). Next, we sought to directly compare the effects of flashes and time on behavioral performance (% correct). First, looking across trials with identical differences in flash number (fixed $|\#R-\#L| = \Delta F$) but with varying total flash number ($\#R+\#L=\Sigma F$) , we calculated the fraction of correct responses as a function of $\Sigma F$, relative to the average performance ($\Delta$ Performance; *Figure 2C*). With the difference in flashes $\Delta F$ thus controlled for, we found that trials with greater numbers of flashes showed a substantial decrease in performance. Linear regression suggested that each additional flash decreased performance by 1.14% (+/-0.1%). With an average of 4 flashes presented per second, increasing total flashes at a fixed flash difference thus produced an average decrement in performance of 4.56% per second. Then, to estimate the effect of time on performance, we calculated $\Delta$ performance across trials with both identical flash differences ($\Delta F$) and identical total flash number $\Sigma F$), but with different overall trial duration. This analysis, which controls

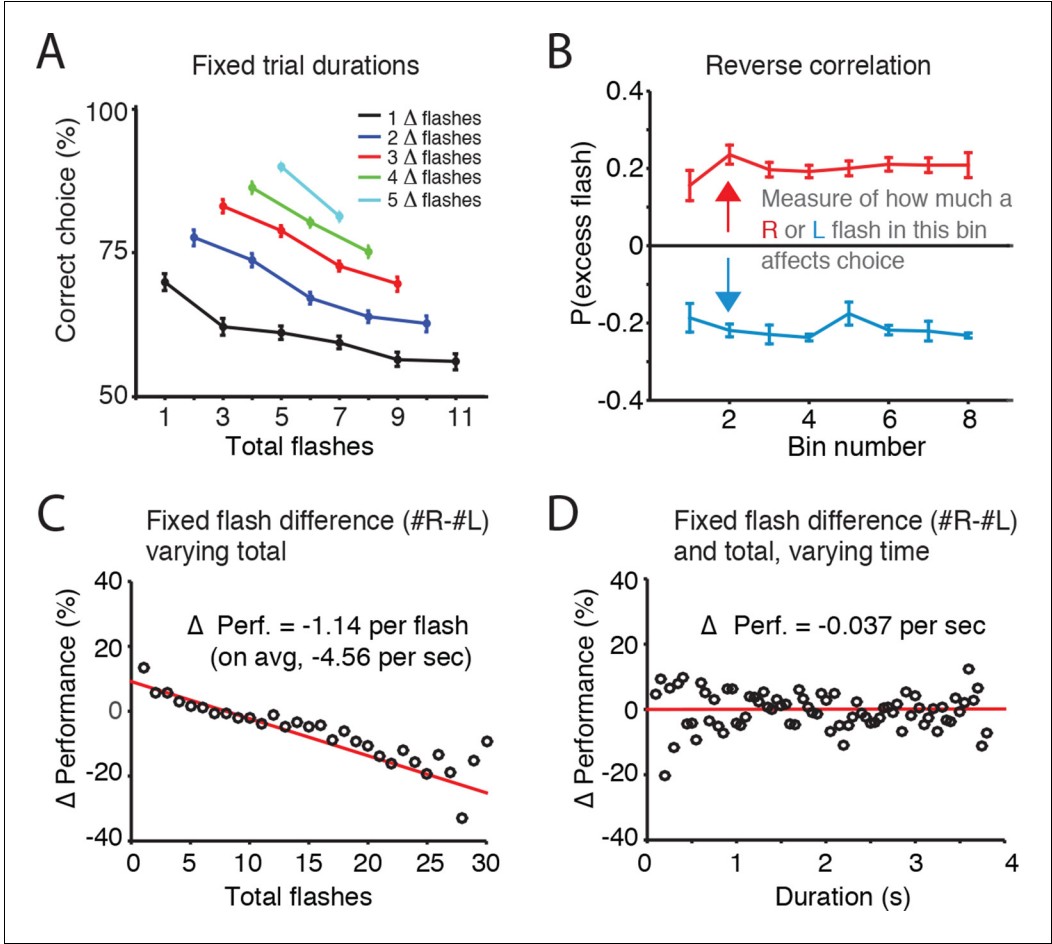

**Figure 2.** Error rate increases with number of flashes, not trial duration. (**A**) For trials of fixed duration and fixed difference in number of flashes (Δ-flashes, colored lines), behavioral performance decreased with more flashes. (**B**) Reverse correlation analysis indicating the relative contribution of flashes occurring at different times in the trial to the subject's behavioral choice. Each point on the upper line represents the probability that on trials in which the subject poked to the right, there was an extra right flash in each time bin. The lower lines represent the same analysis for trials in which the subject poked left. The flatness of the lines suggests that rats use early, middle and late flashes equally to guide their decision. Lines and error bars represent mean and standard error across rats. This result suggests a long time constant of accumulation. (**C**) Changes in behavioral performance (% correct) as a function of the number of total flashes presented. Data points indicate behavioral performance relative to the average performance (Δ Performance) across trials with identical differences in flash number (|#R-#L| = ΔF) but with varied total flash number (#R+#L=F). Red lines are regression lines to those points, weighted by the number of behavioral trials contributing to each point. (**D**) Changes in behavioral performance as a function of the trial duration. Performance across all delay durations was computed for each unique combination of total number of flashes and difference in the number of flashes. For each unique combination of flash number and difference, performance relative to that average performance (Δ Performance) was computed for different trial durations binned in 50 ms bins (black circles). Red lines are regression lines to those points, weighted by the number of behavioral trials contributing to each point.

The following figure supplements are available for figure 2:

**Figure supplement 1.** Fits of drift diffusion-like model to individual rats.

**Figure supplement 2.** Effect of flash number on performance of individual rats.

**Figure supplement 3.** Effect of trial duration on performance of individual rats.

**Figure supplement 4.** Psychophysical reverse correlation for each individual rat.

for the number of flashes, revealed that the purely time-dependent decrement in performance was only 0.037% (+/-. 77%) for each second of trial duration. These results suggest that error rates depend far more strongly on the number of flashes than on trial duration. We found this both for the group data of *Figure 2* and for individual rats as well (*Figure 2—figure supplement 2*; *Figure 2—figure supplement 3*).

To assess whether the rats exhibited long accumulation time constants we next computed the 'psychophysical reverse correlation' (*Brunton et al., 2013*; *Kiani et al., 2008*; *Nienborg and Cumming, 2009* ; see Materials and methods). For each time bin, we computed the probability that there was an excess flash in that bin on the side to which the subjects subsequently oriented. This analysis indicated that flashes across all time bins contribute equally to subjects' decisions, both when data were pooled across rats (*Figure 2B*), and for individual rats (*Figure 2—figure supplement 4*). This result suggests accumulation time constants that are longer than the trial duration, since subjects' choices were equally influenced by early, middle, and late flashes.

Taken together, behavioral analyses and behavioral model fits suggested that rats based their choices on evidence accumulated over the entire trial duration and that the noise associated with each flash was the predominant source of noise in the accumulation process.

## Subjects' estimates of flash number exhibit scalar variability

As described in the introduction, in the drift diffusion framework assumed in previous behavioral models of evidence accumulation, the standard deviation of the total flash-associated noise scales as the square root of the total number of flashes (*Brunton et al., 2013*). However, two popular alternative models, called scalar variability and subitizing, have been proposed in which noise scales differently with evidence. In the scalar variability model, which has been used to describe both time estimation and counting, the standard deviation of the estimate shows a linear relationship with the quantity represented. The subitizing model predicts that the numerical representation of the first few numbers, up to three or four, is essentially noiseless, and after five its standard deviation grows linearly with the number of flashes. Subitizing has been proposed in tasks in which stimuli are presented simultaneously (*Trick and Pylyshyn, 1994*) and also sequentially (*Camos and Tillmann, 2008*; *von Glasersfeld, 1982*).

To quantify the amount of noise associated with different numbers of flashes, we used a signal detection theory-based framework that abstracted away the sequential nature of the stimulus presentation, and focused instead on the total number of flashes on each side, and noise associated with those totals, as the key determinants of the animal's decisions (*Figure 3A*). In this approach, the subjects' estimate of the number of flashes on one side is modeled as a random variable drawn from a Gaussian distribution. For *n* flashes presented on a side, the mean of this distribution is equal to *n*, and the variance is a free parameter, $\sigma_n^2$. If there can be up to 15 flashes presented on a side, then, there will be 16 free parameters in the model, $\sigma_0^2$ through $\sigma_{15}^2$. On each trial the subject selects two such random variables, one for the total number of flashes on the left and one for total flashes on the right, compares the two variables and orients to the side for which the random sample was greater. Correct responses occur when the random sample associated with the side that had the greater number of flashes is greater than the random sample from the other side.

We used maximum likelihood estimation to calculate the standard deviations of each distribution ($\sigma_0 \ldots \sigma_{15}$) given the number of flashes and behavioral choices across all trials. We found that the best-fit standard deviation values scaled approximately linearly with the number of flashes (*Figure 3B*; *Figure 3—figure supplement 1*). We next compared the predictions of the model with the behavioral data (*Figure 3C and D*). The model was able to capture a large number of features of the data, including the subjects' imperfect performance on the easiest trials, even though no lapse rate parameter was included in the model (*Figure 3D*). This was true both for the group data in *Figure 3* and for individual rats (*Figure 3—figure supplement 2*). Note, although our model represents numerical estimates as a scalar random variable on each trial, our results are also consistent with probabilistic numerical representations in the brain (*Kanitscheider et al., 2015*).

We used least squares regression combined with bootstrapping to estimate the best-fit scaling of $\sigma_n$ as a function of *n* according to each of the three models of noise: scalar variability (SV, linear scaling), subitizing (SUB +SV, constant at zero up to some value of *n*, then linear scaling) and linear variance (LV, square root scaling) (*Figure 3E*). Comparison of goodness-of-fit value ($r^2$) using a

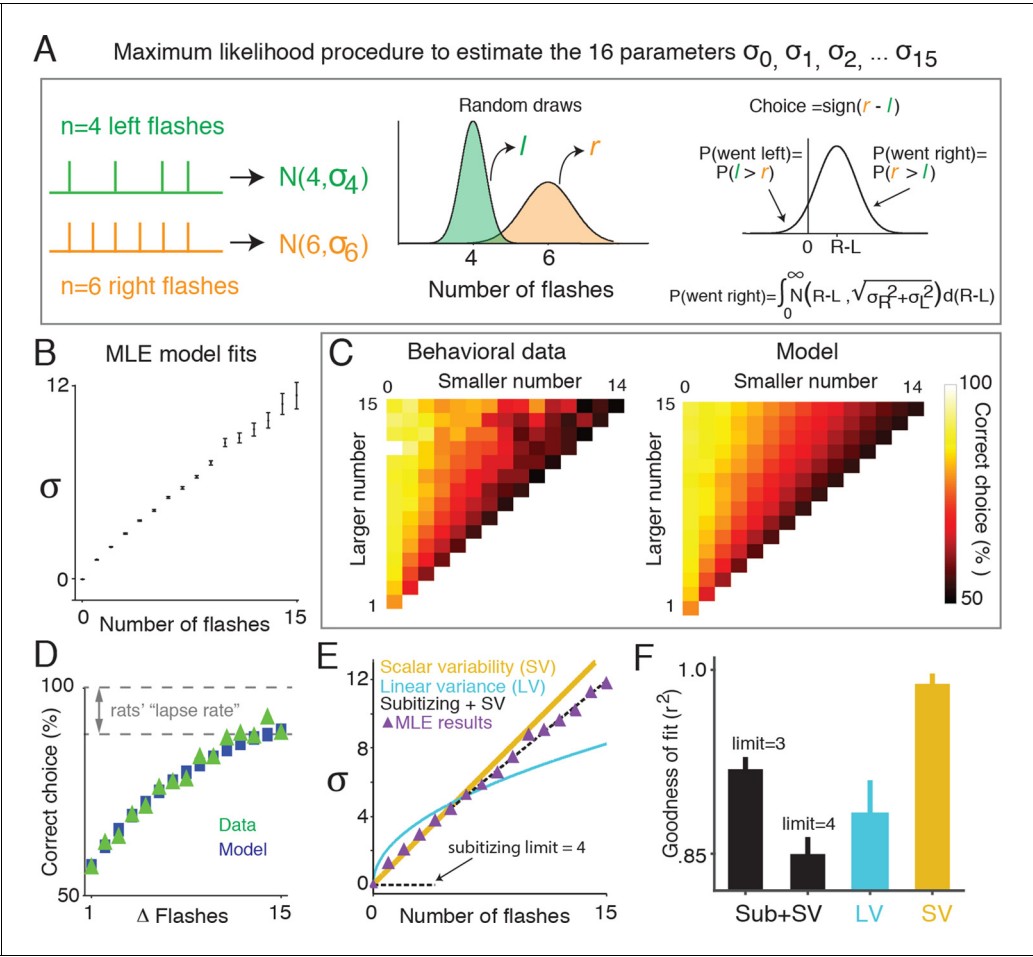

**Figure 3.** Signal detection theory-based model reveals linear scaling of standard deviation of numerical estimates. (A) Schematic of the model used to determine the standard deviation ($\sigma$) of the subjects' estimate of flash number. Left panel indicates the stimuli from an example trial in which four flashes were presented to the left side (green) and six flashes were presented to the right (orange). The model assumes that on any given trial, a subjects' estimate of flash number on each side is a continuous random variable drawn from a Gaussian distribution whose mean is the number of flashes on that side (*n*), and whose variance is a free parameter ($\sigma_n^2$) (*middle panel*). The choice on each trial is determined by comparing these two random variables. Errors occur when the difference of the two random variables (greater magnitude – lesser magnitude) is less than zero. The variance for each Gaussian representing a given flash number ($\sigma_0^2 \ldots \sigma_{15}^2$) was fit to the behavioral data (*right panel*) using maximum likelihood estimation. (B) Model fits of the standard deviations ($\sigma_0 \ldots \sigma_{15}$) in the rats' estimate for different numbers of flashes. Error bars indicate the 95% confidence intervals for the mean based on one thousand-fold resampled data. Note the deviation from pure linear dependence of the $\sigma_n$ parameters on *n* for *n*<2. (C) Comparison of the behavioral data (*left panel*) with the predictions of the model (*right panel*) based on the $\sigma_n$ values calculated as shown in *Figure 3B*. Color indicates the percentage of trials on which the subject responded correctly. (D) Comparison of psychometric performance of the rats (data, green triangles) and model prediction of performance (model, blue squares). (E) Three models that predict how the standard deviation ($\sigma$) of the numerical estimate scales with the number of flashes. Scalar variability predicts that $\sigma$ scales linearly with the number of flashes (SV, yellow). Subitizing predicts that $\sigma$ is zero until a limit (3 or 4), and then follows scalar variability prediction (black dashed). The drift diffusion models predict that the variance of the estimate scales linearly, and $\sigma$ scales with the square root of the number of flashes (LV, blue). Purple triangles are model estimates of $\sigma$, replotted from *Figure 3B*. Each model was fit using linear regression to the model estimate of $\sigma$, weighted by the number of data points contributing to each triangle. Additionally, all models were constrained to intersect with the origin. (F) Goodness of fit of the $\sigma_s$ shown in *Figure 3B* to subitizing (SUB+SV, black), the drift diffusion model (LV, blue) and scalar variability (SV, yellow) using least squares regression. Analysis indicates that the data is best fit by a scalar variability model. Error bars represent the 95% confidence intervals based on fits derived from a thousand-fold resampling of the data.

The following figure supplements are available for figure 3:

**Figure supplement 1.** Signal detection theory-based model fit to individual rats.

**Figure supplement 2.** SDT model prediction vs. data for each rat.

**Figure supplement 3.** Behavior and SDT model approximate scalar variability.

*Figure 3 continued on next page*

*Figure 3 continued*

**Figure supplement 4.** Signal detection theory-based model fit to the auditory (clicks) data.

**Figure supplement 5.** Permutation test comparing goodness-of-fit of scalar variability and linear variance to the auditory (clicks) data.

**Figure supplement 6.** Chronometric plots for the auditory (clicks) data and predictions of scalar variability.

**Figure supplement 7.** Chronometric plots for the auditory (clicks) data and predictions of scalar variability with an offset.

nonparametric permutation test confirmed that the MLE standard deviation estimates ($\sigma_0 \ldots \sigma_{15}$) were better fit by the scalar variability model than the subitizing or linear variance models (*Figure 3F*; see Materials and methods: Model Comparison).

Scalar variability suggests that scaling the number of flashes on each side by the same factor should lead to identical discriminability: if $\sigma_n = k \times n$, then the probability of choosing right (equation in *Figure 3A*) is a function of the ratio r = $N_R/N_L$, where $N_R$ and $N_L$ are the number of flashes on right and left, respectively. For the easiest trials (e.g., $N_L \gg 1$ and $N_R = 0$), performance asymptotes at

$$\frac{1}{\sqrt{2\pi}} \int_{\frac{1}{k}}^{\infty} e^{-x^2/2} dx$$

producing a non-zero lapse rate even without an explicit lapse rate parameter (see mathematical appendix, Appendix 1). To test the prediction that performance should be constant for trials with a fixed ratio r, we compared behavioral performance on trials with constant ratios of flashes on the rewarded and unrewarded sides. Supporting scalar variability, both the data and best-fit signal detection theory-based model predictions exhibited roughly constant performance for trials with fixed ratios of flashes (*Figure 3—figure supplement 3*). Interestingly, however, behavior deviated from scalar variability, especially for trials with very low number of flashes. For these trials, there was an improvement in performance with additional flashes, even though the ratio of flashes was fixed (*Figure 3—figure supplement 3*), which is not predicted by pure scalar variability. Nevertheless, this deviation from scalar variability was well predicted by the model parameters of *Figure 3B*. It is thus likely driven by the deviation from pure linear dependence of the $\sigma_n$ parameters on $n$ for $n<2$ (*Figure 3B*).

We repeated the signal detection theory-based analysis on behavioral data collected from rats performing a previously developed auditory accumulation of evidence task (*Brunton et al., 2013*; *Hanks et al., 2015*). We accounted for adaptation effects, which are prominent in the auditory task due to high click rates, and computed the 'effective' number of clicks presented after implementing the adaptation dynamics described in *Equation (6)* (see Materials and methods). On trials in which this approach yielded fractional number of clicks, we rounded the effective click number (after adaptation) to the nearest integer. This kept the number of parameters in the auditory task model roughly equal to the number in the model for the visual task. We then used maximum likelihood estimation to calculate the standard deviations of each distribution ($\sigma_0, \ldots, \sigma_n$) given the effective number of clicks and the subjects' behavioral choices. Consistent with our results with the visual flashes task, the goodness-of-fit value ($r^2$) revealed that the MLE standard deviation estimates in the auditory task were better fit by a model of scalar variability plus a constant offset (SV, $\sigma_n = k_0 + k \times n$) compared to the linear variance previously assumed by sequential sampling models (LV, $\sigma_n^2 = k_0 + k \times n$) ($r^2 = 0.97$ for SV vs $r^2 = 0.89$ for LV, p << 0.001 based on nonparametric permutation test, *Figure 3—figure supplement 4*, *5*). Our findings are thus not specific to the visual modality. Notably, perfect scalar variability ($\sigma_n = k \times n$ without the constant $k_0$) would predict no improvement as a function of time for a fixed click ratio, yet such an improvement is clearly seen in the clicks data (*Figure 3—figure supplement 6*). The SV model accounted for this by having a significantly non-zero offset, $k_0$, and was thus able to fit the clicks data quite well (*Figure 3—figure supplement 7*). A non-zero fixed offset, $k_0$, thus represents a source of noise that is added to the accumulators, independently of the stimulus that was presented. Notice that this source of noise does not account

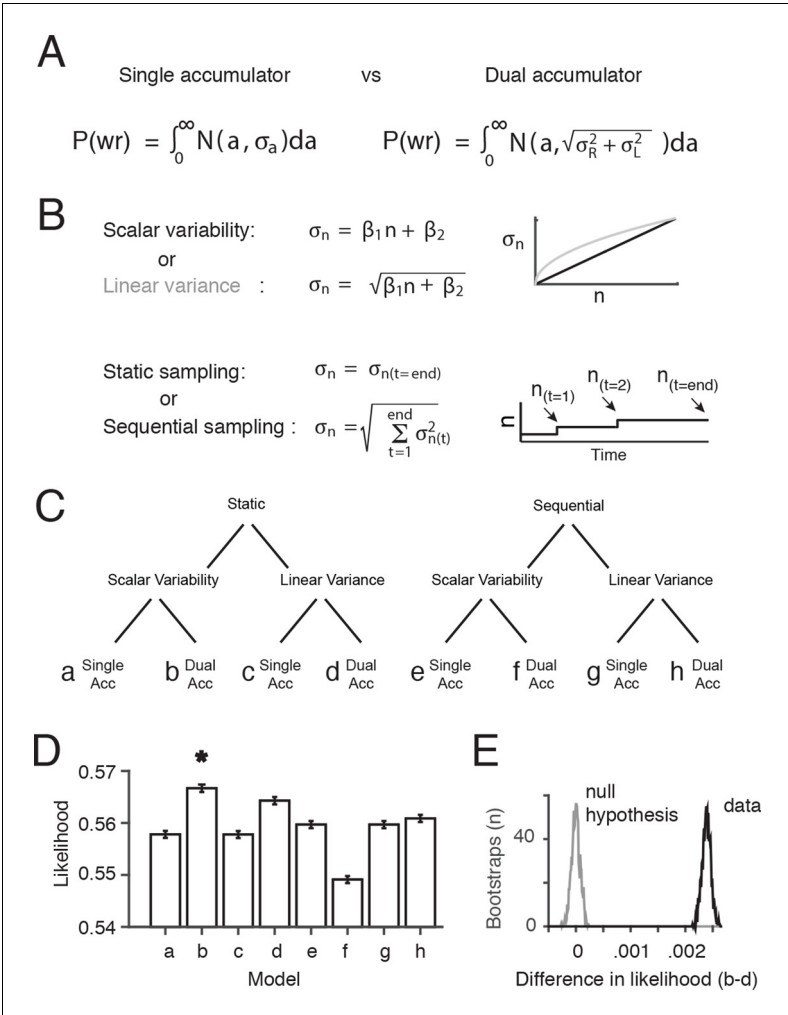

**Figure 4.** Comparison of behavioral models suggests the presence of at least two accumulators. (**A**) General forms of the signal detection theory-based models that were compared assumed either single or dual accumulators. Each model determines which choice to make on each trial, by randomly selecting a value (*a*) from a Gaussian with mean equal to the difference in total right minus left flashes, and standard deviation $\sigma_n$. If *a*>0, the model decides "Right", and if *a*<0 the model decides "Left". Noise, parameterized as the standard deviation of the distributions of flash number or flash difference ($\sigma_n$), enters the accumulation process differently in each model. Single accumulator models assume that noise depends only on the difference in the number of flashes (R-L) while dual accumulator models assume that noise depends on the total number of flashes (R+L). (**B**) For each class of models (single vs. dual accumulators), we implemented versions that assumed scalar variability or linear variance and that assumed a static or a sequential sampling process. For the static model, equivalent to our original signal detection theory-based model, noise scales with the number of flashes seen at the end of the trial. For the sequential sampling model, noise is added to the subject's estimate with each flash, and the magnitude of the noise depends on the value of the accumulator at the time of the flash (*a(t)*). (**C**) Cladogram indicating the details of each of the eight models (a-h). (**D**) Comparison of the likelihood of the model given the data for each of the eight model versions. A bootstrapping procedure was used, in which behavioral trials were resampled (with replacement) and each model was fit to each resample. Error bars indicate the 95[th] percentiles of the model likelihoods using the best-fit parameters for all resamples. The number of parameters is equal across all models, allowing direct comparison of likelihoods. (**E**) The permutation test used to assess statistical significance. Black line indicates the distribution of the differences of likelihoods between the two models with the largest likelihood, scalar variability, dual accumulator static version (**b**) and linear variance, dual accumulator, static version (**d**), across all resamples. Gray line indicates the distribution under the null hypothesis (see Materials and methods: Model comparison). Model b has a significantly larger likelihood than model d and all other models (p<<.001; see Materials and methods: Model comparison).
*Figure 4 continued on next page*

*Figure 4 continued*

The following figure supplements are available for figure 4:

**Figure supplement 1.** Model comparisons for each rat.

**Figure supplement 2.** Model predictions versus data.

for the lapse rate, since $k_0$ does not appear in the expression for the lapse rate (see mathematical appendix, Appendix 1). A larger offset ($k_0$) was the main qualitative difference we found between fitting the auditory clicks data (for which $k_0$ = 2.10 +/- 0.20, mean +/ s.e.m. across rats) and the visual flashes data (for which $k_0$ = 0.49 +/- 0.142).

## Comparison of behavioral models suggests the presence of at least two accumulators

The signal detection theory-based model described above makes two assumptions about the nature of the accumulation process. First it assumes that subjects maintain separate estimates of left and right flashes during a behavioral trial. Second it assumes that noise depends only on the final value of the left and right integrator. However, many quantitative models of accumulation of evidence propose that subjects maintain a single decision variable (e.g., a running estimate of the difference in number of right minus left pulses; *Bogacz et al., 2006*). Other binary decision models have proposed two separate accumulators in which noise is added throughout the accumulation process (*Ratcliff et al., 2007*; *Usher and McClelland, 2001*).

Therefore, we decided to evaluate the assumption of a dual accumulator and the assumption that the noise depends only on the final value of the integrator(s), by comparing the performance of multiple variants of our signal detection theory-based behavioral model (*Figure 4*). We probed three independent questions, the combined answers to which produced eight different models: (1) Is performance best fit by dual accumulators or by a single accumulator? (2) Is performance best fit by gradual accrual of noise throughout stimulus presentation, or by noise that depends only on the final value of the integrator? And finally, as in the previous section, (3) Is variability best fit as standard deviation being linear in flash number, or by variance being linear in flash number? Each model (a-h) had two free parameters $\beta_1$ and $\beta_2$ that determined the relationship between the number of flashes, n, and the standard deviation of the flash number, $\sigma_n$ (*Figure 4B*). Four models (a,b,e,f) assumed scalar variability ($\sigma = \beta_1 * n + \beta_2$) and four (c,d,g,h) assumed linear variance ($\sigma^2 = \beta_1 * n + \beta_2$). Four models assumed a single accumulator (a,c,e,g) and four assumed independent left and right accumulators (b,d,f,h). Four models (a-d) assumed that noise was based only on the final estimate of the accumulator (static sampling), and four assumed (e-h) that noise was added to the estimate of the accumulator at the time of each flash (sequential sampling).

We used maximum likelihood estimation to find, for each model, the values of $\beta_1$ and $\beta_2$ that best fit the behavioral data. Despite the closely related structure of the eight models, and having precisely the same number of free parameters, the likelihood of the data at the best-fit parameter values for nearly all of the two-accumulator versions (b,d,h) of the model was significantly greater than the corresponding one-accumulator versions (a,c,g) (p<.001; nonparametric test using bootstrapping; see Materials and methods), indicating better performance for the two-accumulator model (*Figure 4D,E*). Overall we found that the static, dual accumulator scalar variability model had the highest likelihood across all models (p<.001; nonparametric test using bootstrapping see Materials and methods: Model Comparison; *Figure 4—figure supplements 1*, *2*). This suggests that the main noise source, in the limited regime of models considered here, operates on the final value of the accumulators, rather than being incrementally and sequentially added to the accumulators as the process unfolds. We emphasize, however, that we have not made an effort to systematically explore the full space of possible models in which noise is added gradually during the stimulus presentation, so a conclusion that noise is added only to the final value of the accumulator remains tentative.

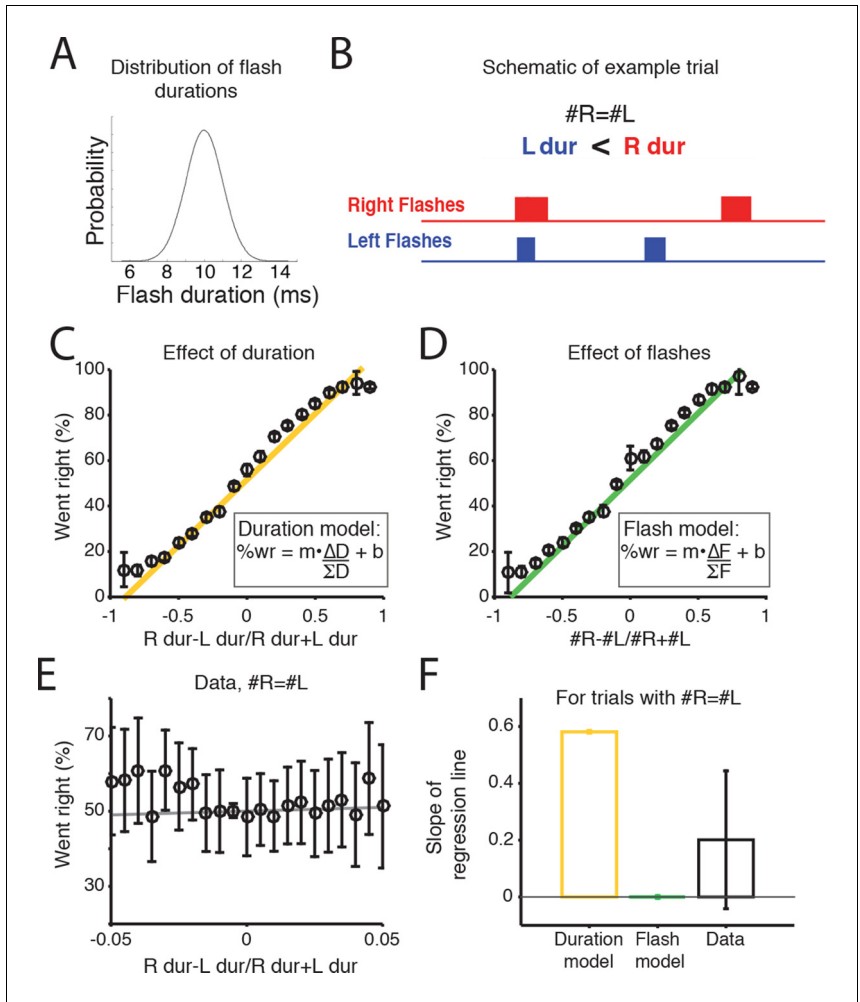

**Figure 5.** Rats accumulate flash number, not duration. (**A**) Distribution of flash durations. In a subset of experiments, flash durations were drawn from a Gaussian distribution with a mean of 10 ms. As a consequence, on some trials in which the same number of flashes were presented to both sides (#R=#L), there was a greater overall flash duration on one side (up to 50 ms) (**B**) Schematic of the timing and duration of left (blue) and right (red) flashes on an example trial in which #R=#L, but there was greater overall flash duration from the right LED. (**C**) Percent of trials on which the animal went right as a function of difference in right-left LED duration (specifically (durR-durL)/(durR+durL)) across all combinations of left and right flash number. Because LED duration is correlated with number of flashes on most trials, this looks very similar to performance as a function of flash number. Yellow line is the regression line to the data. (**D**) Percent of trials on which the animal went right as a function of difference in flash number ((#R-#L)/(#R+#L)). Green line is the regression line to the data. (**E**) Behavioral choice as a function of the difference in the overall flash duration ((durR-durL)/(durR+durL)) on trials with equal numbers of flashes on both sides (#R=#L). Black circles indicate the percent of trials in which animals went right on trials of different overall flash durations. Error bars indicate 95% confidence intervals for the mean assuming a binomial distribution. Solid gray line is the regression line to the data, whose slope is displayed in panel F. (**F**) Bar plots indicate the slope of the regression lines in panels C and D for trials with equal numbers of flashes. Error bars are standard error of the coefficient (slope) estimates. These represent different models of what the slope of the regression line to the data would be if the animal were integrating flash duration ('Duration model', yellow) or flash number ('Flash model', green). The slope of the regression line fit to the data ('Data') is significantly different from the line relating the LED duration to choice, but not significantly different from zero. This suggests that rats integrate flash number.

## Subjects accumulate flash number, not flash duration

Scalar variability has been proposed both for numerical and interval timing estimation, raising the possibility that rats could be accumulating the summed duration of the flash 'ON' times. Alternatively, it is possible that rats could be treating each flash as a distinct event, independent of its duration, and integrating the number of flashes. To test which strategy the rats used, we performed an additional experiment in which we trained a second cohort of rats (n=7) on a new version of the task. As before, rats were rewarded for orienting to the side with the greater number of flashes, but in this version the durations of each flash and inter-flash interval were randomly jittered (see Materials and methods). Flash duration was drawn from a Gaussian distribution with a mean of 10 ms and standard deviation of 1.5 ms (*Figure 5A,B*). Because LED duration is correlated with number of flashes on most trials, it is difficult to evaluate whether animals use flash duration or flash number on most trials (*Figure 5C,D*). However, on trials in which there was no difference in the number of flashes ($\Delta$flashes = 0), the jittering of flash duration led to the generation of some trials with a difference in the duration presented from the right or left LEDs (*Figure 5B*). Since, for example, 20 ms of light is the equivalent of a difference of two flashes, we reasoned that on these trials, if the subjects were integrating light duration, they would be more likely to orient to the side with longer LED durations, and a regression line fit to the data in *Figure 5E* should have a positive slope. Conversely, if they were integrating discrete flash number, they should not exhibit a preference to the side with longer flash duration, and a regression line fit to the data in *Figure 5E* should have a slope of zero. Consistent with integrating flash number, we found that on trials with 0 $\Delta$flashes, rats did not exhibit a preference for the side with longer flash durations (*Figure 5E,F*). These results indicate that the rats were accumulating the number of flashes, not their total duration.

## Reward and error history bias future behavior choices

One potential additional source of variability in the rats' behavior is the location of reward on previous trials. In some behavioral tasks, subjects display a win-stay-lose-switch approach to decision-making, in which choices that lead to reward tend to be repeated, while alternative unrewarded choices tend to be abandoned. To assess whether the memory of reward location on previous trials biased the subjects' decisions on future trials, we identified trials following a reward and compared behavioral performance depending on whether the reward had been delivered to the left or right (*Figure 6A*). On these trials subjects continued to perform the accumulation task, but were biased toward the side where they had previously obtained reward. The effect of this bias was modest but significant, and was constant across a range of numerical flash differences, producing a vertical shift in the psychometric function (*Figure 6A*). Similarly, if the subjects made an error on one side, they exhibited a bias toward the other side on the subsequent trial (*Figure 6B*).

Next, we extended this analysis to determine how long this bias persists. We computed the probability that subjects chose to return to the same side where they had previously obtained a reward and repeated this computation for progressively later choices (*Figure 6C*, black line). The behavioral side bias caused by a reward decreased steadily over the next three trials and after three trials no significant bias was observed. We repeated this analysis for error trials and found that the bias was smaller in magnitude, but displayed the same steady decrease for three trials into the future (*Figure 6C*, gray line). After three trials no significant bias was observed.

Finally, when rewards were consecutive (i.e. more than two correct responses to the right in a row) the bias observed was additive. The side bias observed from two consecutive rewards on the same side is not significantly different from the sum of the reward biases from one and two trials back (*Figure 6D*). Together these results suggest that (1) both rewards and errors bias the subject's choices up to three trials in the future and (2) reward history effects combine linearly to influence the behavior of the subject. Similar results have been observed in the auditory, Brunton et al., version of the accumulation of evidence task (Chan, Brunton, and Brody, unpublished data).

## Performance of voluntarily head restrained rats on a visual accumulation of evidence task

Future studies into the biological mechanisms of decision-making will be facilitated by the ability to perform precise neuronal circuit perturbations and cellular resolution imaging. Behavioral tasks that can be performed during head restraint allow the experimenter to perform techniques such as

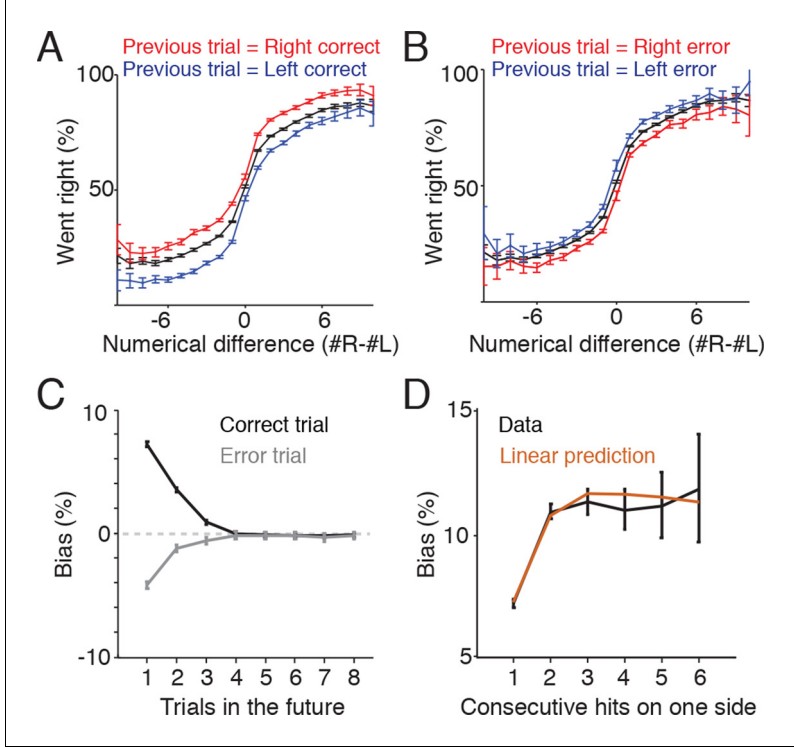

**Figure 6.** Trial history contributes to behavioral variability. (**A**) Reward biases decision on subsequent trials. Plot indicates psychometric performance on trials following correct right (red) and correct left (blue) trials. Black line indicates the mean. Data is pooled across all sessions for all rats. Error bars are 95% confidence intervals for a binomial distribution. (**B**) Psychometric performance on trials following errors. Making an error on one side modestly decreased the probability that subjects would orient to that side on the subsequent trial. (**C**) Following a rewarded trial, subjects exhibited increased probability of returning to the same side poke up to three trials in the future (black line). Following an error trial, they exhibited a decreased probability of orienting to the same side up to three trials in the future (gray). (**D**) Effects of reward history for correct trials are additive. Black line indicates the probability of returning to the same side given 1-6 consecutive rewards on the side. Brown line indicates the bias predicted from the linear sum of the biases observed for non-consecutive rewards as shown in *Figure 6C*.

cellular resolution optical stimulation and large-scale two-photon calcium imaging that are difficult to employ in unrestrained preparations. Therefore we developed a version of the visual accumulation of evidence task that could be performed during voluntary head-restraint.

In the head-restrained version of the task, subjects initiate a behavioral trial by guiding a surgically implanted titanium headplate into a custom headport mounted on one wall of an operant training chamber (*Figure 7A*, *left* and *7B*). The trial begins when the leading edge of the headplate contacts two miniature snap action switches (contact sensors) mounted on the headport. Voltage-controlled pneumatic pistons then deploy to immobilize the headplate and, using the principles of kinematic mounts, register the head to within a few microns. During the restraint period, which lasts 2–3 s, head-restrained animals are presented with up to six flashes on each side (*Figure 7A*, *middle*). After a brief (500 ms) delay, subjects are released from restraint, and can obtain water reward by orienting to one of two side ports mounted to the wall flanking the headport (*Figure 7A*, *right*). Head-restrained subjects can also choose to terminate the restraint period early by operating a release switch located on the floor of the chamber. Trials that are terminated early result in a brief timeout (~2 s) in which no reward can be obtained and no trial can be initiated.

To compare behavioral performance to unrestrained rats, we trained seven rats on the head-restrained version of this task. The initial stages of training were performed in a high-throughput facility, while acclimation to the pistons and data collection from fully trained animals were performed in a separate facility with a single dedicated behavioral chamber. Rats completed the initial stages of training in similar time to the unrestrained rats. However, a portion of rats (~40%) were

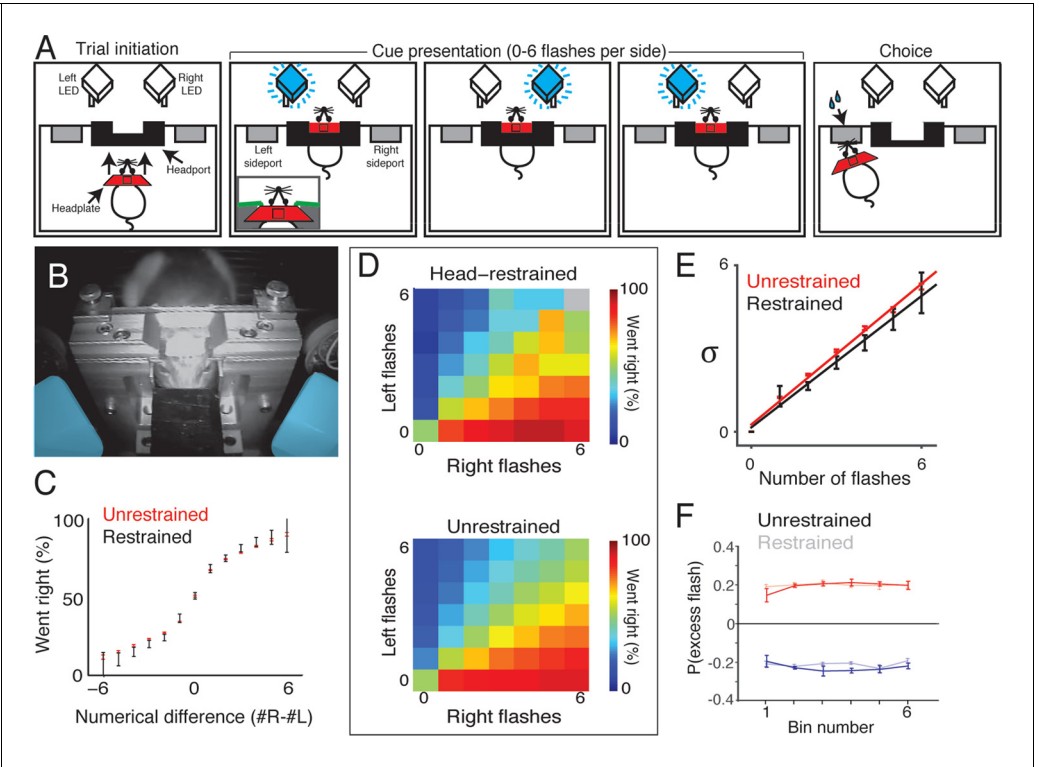

**Figure 7.** Head-restrained and unrestrained rats exhibit comparable task performance. (**A**) Schematic of the accumulation of evidence task during head restraint. A rat initiates a behavioral trial by inserting his headplate into a custom headport along one wall of an operant training box (*left panel*). While voluntarily head restrained, the rat is presented with a series of pseudo-randomly timed flashes from blue LEDs to the left and right in front of him (*middle panels*). Following flash presentation, the subject is released from restraint and is free to withdraw his head from the headport and orient to one of the side ports to obtain reward (*right panel*). (**B**) Image of a rat training on voluntary head restraint. The left and right stimulus LEDs, which are mounted in light diffusing boxes, are pseudo-colored in blue. (**C–F**) Direct comparison of behavioral performance between seven head restrained rats and seven unrestrained rats. (**C**) Comparison of psychophysical performance indicates similar sensitivity to the difference in the number of flashes between restrained (black) and unrestrained rats (red). Error bars indicate 95% binomial confidence intervals on the data. (**D**) Comparison of behavioral performance between restrained (upper panel) and unrestrained (lower panel) rats across all different stimulus types in the restrained version of the task. Color indicates the percentage of trials on which the subjects oriented to the right. Gray indicates no data. (**E**) Comparison of the noise (standard deviation) in the estimates of the number of flashes for restrained rats (black) and unrestrained rats (red) using the procedure described in *Figure 3*. Lines indicate the best linear fit for the restrained (black) and unrestrained (red) cohorts. Error bars indicate the 95% estimates of the means of $\sigma_n$. (**F**) Reverse correlation analysis indicating the relative contribution of flashes occurring at different times in the trial to the subjects' behavioral choice for restrained (transparent) and unrestrained (solid) rats. Data from unrestrained rats are a subset of the data plotted in *Figure 2F* (color convention is also adopted from *Figure 2F*). Lines and error bars are mean and standard error across rats.

slower to acclimate to the pistons than the remainder of their cohort and were excluded from further study.

Behavioral analysis revealed comparable performance between voluntarily head-restrained and unrestrained rats (*Figure 7C and D*). Fitting the accumulator model produced model parameters in the head-restrained subjects that was similar to the best fit parameters for the unrestrained rats: negligible accumulator noise, predominant flash-associated noise, accumulation time constants longer than the trial duration and bounds higher than the number of possible flashes. Fitting the signal detection theory-based model revealed similar scaling of noise between restrained and unrestrained versions of the task (*Figure 7E*). We computed the 'psychophysical reverse correlation' to further assess whether the rats exhibited long accumulation time constants (see Materials and methods). For each time bin, we computed the probability that there was an excess flash in that bin on the side to which the subjects subsequently oriented. This analysis indicated that flashes in each time bin contribute equally to subjects' decisions (*Figure 7F*). This was true for both voluntarily head-restrained and unrestrained rats.

## Discussion

In this study we developed a visual accumulation of evidence task to study the sources of noise that contribute to behavioral variability. Subjects were presented with two simultaneous trains of randomly timed flashes, one train on their left, and the other on their right, and were rewarded for orienting towards the side with the greater total number of flashes. Behavioral analyses, including reverse correlation, indicated that rats used flashes presented throughout the entire trial to guide their decision, consistent with a strategy of accumulating evidence (*Figure 2—figure supplement 4*). We used a signal detection theory framework to estimate the noise associated with each pulse of evidence (flash). This analysis showed that noise (standard deviation) in the subjects' numerical estimates scaled linearly with the number of flashes presented, indicating that noise per flash is not independent across flashes. We find that noise is also independent of total stimulus duration, and is perhaps added to the decision variable after the accumulation process. Our behavioral modeling further suggests the existence of two separate accumulators in this binary decision task (*Figure 4*). The noise scaling we found is similar to scalar variability described in the numerical cognition literature, and appears to be the major source of variability in our subjects' behavior. Yet it is not diffusion-like and is therefore not well-described by the majority of current 'drift-diffusion modeling' (DDM) approaches for accumulation of evidence in decision-making. Incorporating this scalar variability noise will likely be critical for understanding variability in decision-making behaviors. Finally, we demonstrate that rats can be trained to perform this task during voluntary head restraint, which opens up new experimental avenues for uncovering the circuit mechanisms that generate behavioral variability during decision-making.

### Performance of cognitive behaviors during voluntary head restraint

Head-restraint is an experimental tool widely used in psychology and neuroscience to immobilize the head of a subject. The technique is typically employed in behavioral experiments to facilitate reliable presentation of sensory stimuli to the head or face and precisely control the subjects' movements, and in neurophysiological experiments to minimize motion between the brain and recording apparatus. However, it can take some species, notably rats and primates, weeks to acclimate to forced head restraint. Moreover forced head restraint requires skilled experimenter intervention and is difficult to incorporate into automated training procedures. Voluntary head restraint has been proposed to reduce the stress normally associated with forced head restraint and facilitate training on sophisticated behavioral tasks (*Scott et al., 2013*). Here, we show that rats are able to perform a complex cognitive task, involving both numerical representation and integration of evidence, while voluntarily head-restrained. This provides a foundation for identifying and characterizing the neural circuit mechanisms of accumulation of evidence and numerical representation in future studies using *in vivo* cellular resolution imaging and perturbation (*Rickgauer et al., 2014*).

### Sources of noise in accumulation of evidence tasks

The behavioral task described here exhibited a number of features that enabled detailed characterization of noise during accumulation of evidence. The visual stimuli presented in this study were: (1) low in number; (2) presented reliably to left and right visual hemifields; (3) presented from well-separated light sources in front of the rat's face; (4) presented at sufficient intervals to avoid adaptation or facilitation between successive stimuli. All of these features attempted to minimize sources of noise other than noise in the subjects' internal estimates. Consistent with previous reports (*Brunton et al., 2013*; *Kiani et al., 2013*), we found that behavioral variability was not affected by the trial duration but was strongly correlated with the number of flashes presented.

However, we identified a number of additional sources of noise. First, rats exhibit a modest side bias due to reward history occurring up to three trials in the past (*Figure 6*). In dynamic foraging tasks, strategies that incorporate trial history (such as win-stay, lose-switch strategies) often maximize rewards (*Corrado et al., 2005*; *Herrnstein, 1961*; *Sugrue et al., 2004*). However, in the visual accumulation of evidence task, the rewarded side is assigned independently on each trial, so trial history adds noise to the decision process. The influence of trial history on subsequent decisions has been observed in a number of other studies, suggesting that it is a general source of behavioral variability that should be incorporated into accumulation of evidence models (*Busse et al., 2011*; *Gold et al., 2008*).

We also found that the standard deviation in the accumulated estimate scaled linearly with the number of flashes. This type of noise scaling, called scalar variability, was consistent across the range of stimuli presented (1-15 flashes, with minor deviations at the lowest number of flashes) suggesting that the subjects did not use a subitizing approach, i.e., they did not use rapid accurate counting for numbers less than five. We applied our analysis to a closely related previous auditory task that, as in our current visual task, also delivered sensory evidence in discrete pulses (*Brunton et al., 2013*). We found that our conclusions also applied to the auditory data, with the standard deviation of the noise after $N$ flashes well-described by $\sigma_n = k_0 + k \times n$. Our observation that performance in the auditory task improves with increased integration times is qualitatively consistent with data from tasks in which evidence is delivered in a continuous stream rather than in discrete pulses. This leads us to suggest that scalar variability should be an important consideration in the analysis of data from continuous tasks. But whether or not scalar variability is the dominant form of noise in continuous stream tasks, as it was in the pulsatile tasks analyzed here, remains to be determined.

## Implications for neural mechanisms of perceptual decision-making

The primary focus of this work has been to provide a quantitative description of behavior during accumulation of evidence and to evaluate the sources of behavioral variability. However, an important distinction should be made between quantitative models of behavior and mechanistic models of the underlying neural processes (*Marr, 1982*). Our results do not suggest an underlying neural mechanism, however they do place important constraints on such mechanistic models. One such constraint is scalar variability, which has been observed in a wide variety of tasks requiring animals to estimate magnitudes (such as time or number *Gallistel and Gelman, 2000*; *Gibbon, 1977*). Two possible neural mechanisms have been proposed to explain this phenomenon. The first is that the brain represents magnitude on a logarithmic scale with constant noise (*Fechner, 1860*; *Dehaene and Changeux, 1993*; *Meck and Church, 1983*; *Nieder and Dehaene, 2009*). Neurophysiological recordings, particularly in sensory systems, have provided support for this mechanism. For example, neurons in the rodent somatosensory cortex exhibit firing rates that scale logarithmically with frequency of tactile stimulation (*Kleinfeld et al., 2006*). In addition, neurons in primate parietal cortex appear to encode numerosity on a nonlinear scale (power law or logarithmic; *Nieder and Miller, 2003*). The second possibility is that the brain represents magnitude on a linear scale with noise that is proportional to the magnitude. The results described here did not allow us to differentiate between these two models, and indeed, both models make identical psychophysical predictions (*Dehaene, 2001*). However, how they would be instantiated in the brain is quite different. For the logarithmic case, the signal diminishes with the quantity being represented, but noise added at each time point is independent. For the linear case, the signal is constant (implying near perfect summation of signals), but noise scales linearly with the magnitude being estimated.

## Relation to accumulation of evidence models

Sequential sampling models of accumulation of evidence represent the decision process as the movement of one or more latent decision variables towards a criterion value. Such models have been useful to study a wide range of decision processes. One key advantage is that they provide a moment-by-moment estimate of the decision variable on each trial. This time-varying estimate can be compared to the observed neuronal dynamics, allowing analysis of the correlation between cellular firing rates and the subject's internal representation of evidence (*Hanks et al., 2015*).

Our results suggest three modifications that could improve the accuracy of sequential sampling models of accumulation of evidence. First we found that reward and error trials introduce a bias that persists up to three trials into the future. Second we found that standard deviation in the decision variable increased linearly with the total number of flashes. Note that this observation is inconsistent with the common assumption made in drift diffusion models, that noise is added independent of time and the value of the decision variable. Third, the data was better fit by models that accumulate left and right flashes separately than by models that accumulate a single decision variable. The existence of two accumulators (*Ratcliff et al., 2007*; *Usher and McClelland, 2001*) has been previously observed in the vertebrate brain, for example the oculomotor system in the goldfish contains two lateralized accumulators that integrate transient motor commands to maintain a memory of gaze

position (*Aksay et al., 2007*), and is consistent with electrophysiological recordings in primates (*Bollimunta and Ditterich, 2012*) and rodents (*Hanks et al., 2015*).

In the future, it will be interesting to develop a reaction-time version of the task, in which subjects are allowed to determine the time during which they observe the stimulus. This would allow testing predictions from recent pulse-accumulation models (*Simen et al., 2015*).

### Noise as a mechanism of generalization during perceptual learning

Variability in behavior and neuronal networks has been proposed to result, in part, from biophysical constraints of the nervous system. In some systems, however, noise is thought to serve an important functional role. For example it has been proposed that random fluctuations in neuronal membrane potential contributes to contrast invariance in the visual system (*Anderson et al., 2000*). Moreover, during motor learning, noise is actively introduced into highly reliable systems to create the behavioral variability required for trial and error learning (*Olveczky et al., 2005*). An interesting question is whether noise also facilitates learning in perceptual tasks. One potential function of noise in the estimate of flash number is that it could allow animals, which have been trained on a reduced stimulus set, to generalize, and perform correctly on trials with numbers of flashes they have never encountered. For example, if rats were only trained on trials with 5 and 6 flashes, because of the variability of their estimate of 5 and 6 flashes, on a portion of those trials they will have effectively perceived 7 and 8 flashes, and could learn the correct response to those trials by trial and error. Thus noise may allow the observer to experience a larger range of stimuli than they actually encountered, enabling generalization.

### Conclusion

High-throughput operant conditioning combined with detailed, quantitative analysis of behavioral variability allowed us to characterize the process of evidence accumulation in rats. This analysis revealed that animals' choices were likely derived from a circuit with multiple accumulators and that decisions were influenced by noise from multiple sources. The major source of noise displayed 'scalar variability' in the sense that its standard deviation scaled linearly with the number of pulses of evidence. This observation is inconsistent with the assumption made by most previous models of accumulation for decision-making, namely that the primary noise source is diffusion-like, added to the accumulator independent of time or accumulator value. Our observation is instead consistent with previous models of numerical cognition, and suggests that evidence accumulation and numerical cognition may be subserved by similar or related circuits and neural mechanisms. We speculate that noise that obeys scalar variability could be an important component in models of accumulation of decision-making evidence generally. Finally, the observation that voluntarily head-restrained rats exhibit similar behavioral performance to unrestrained rats, and can be trained to perform complex decision-making tasks, suggests the possibility of using cellular resolution optical imaging and perturbation technologies to characterize the neural substrates of cognition.

## Materials and methods

### Animal subjects

Animal use procedures were approved by the Princeton University Institutional Animal Care and Use Committee (IACUC; Protocols #1837 and #1853) and carried out in accordance with National Institutes of Health standards. All subjects were male Long-Evans or Sprague Dawley rats weighing between 200 and 500 g (Taconic, NY). Rats were placed on a water schedule in which fluids are provided during behavioral training. If rats consumed less than 3% of their body weight in water, they received ad lib water for 1 hr.

### Behavior

Rats progressed through several stages of an automated training protocol before performing the task as described in the results. All data described in this study were collected from fully trained rats. Sessions with fewer than 100 completed trials were excluded from analyses. These sessions were rare and usually caused by hardware malfunctions. From the sessions included in this paper, on average, rats completed 385 trials per session (median: 362 trials per day; range: 128–1005 trials per

day), and performed at 70% correct (median: 71%; range: 56–88% correct; see *Figure 1—figure supplement 1*).

## Stimulus generation

The cue period consisted of fixed time bins, each 250 ms in duration. Unrestrained rats experienced up to 15 bins, whereas head-fixed rats experienced up to 6 bins. For a given trial, each of the two LEDs had a fixed generative probability of producing a flash in each bin. In the final training stage for the unrestrained rats, the generative probabilities for the high-probability LED and low-probability LED ranged from 70-80% and 20-30%, respectively. For restrained rats, the generative probabilities were 70% and 30%. However, the rats were rewarded based on the number of flashes that were actually presented, not on the underlying generative probabilities. In other words, in the rare event that the LED designated 'high-probability' happened to produce fewer flashes than the LED designated 'low-probability', the animal was rewarded for orienting to the side that had more flashes. The flashes were 10 ms in duration (except for an experimental group in which the flash durations were jittered; see *Figure 5*), and they occurred in the first 10 ms of each cue bin. For the unrestrained rats, the inter-flash intervals ranged from 240 ms (two flashes in subsequent bins) to 3.45 s (two flashes separated by 13 bins; see *Figure 1—figure supplement 1*). For the head-fixed rats, the inter-flash intervals ranged from 240 ms to 1.24 s. In a subset of experiments, inter-flash intervals and flash durations were jittered. In these experiments, flash durations were drawn from a Gaussian distribution with a mean of 10 ms, and standard deviation of 1.5 ms. Inter-flash intervals were drawn from a Gaussian distribution with a mean of 250 ms and standard deviation of either 25 or 50 ms.

In the unrestrained version of the task, there was a 500 ms pre-cue period before the beginning of the cue period. For the majority of trials, the duration of the post-cue (or memory) period was randomly drawn from a uniform distribution from 5 to 500 ms. In a small subset of trials, the post-cue period was drawn from a broader distribution ranging up to 6.5 s, but those trials represent a minority of the dataset (see *Figure 1—figure supplement 1*). In the head-fixed version of the task, the pre-cue period was 1 s in duration, to allow time for the pistons to actuate and fix the headplate in place. The post-cue period was 500 ms.

## Behavioral control system

Operant training chambers were controlled by the freely available, open-source software platform Bcontrol (Erlich et al, 2011; *Scott et al. 2013*). Bcontrol consists of an enhanced finite state machine, instantiated on a computer running a real-time operating system (RTLinux), and capable of state transitions at a rate of 6 kHz, plus a second computer, running custom software written in MATLAB. Each behavioral trial consisted of a sequence of states in which different actuators—for example, opening of a solenoid valve for water reward—could be triggered. Transitions between the states were either governed by elapsed times (e.g., 40 ms for water reward) or by the animal's actions, which caused changes to the voltage output of a sensor in the chamber.

## Surgery

Surgical procedures for headplate implantation have been previously described (*Scott et al., 2013*). Briefly, we anesthetized animals with isoflurane in oxygen and gave Buprenorphine as an analgesic. Once anesthetized, the scalp and periosteum were retracted, exposing the skull. Dental cement (Metabond) was used to bond the headplate to the skull. After a 2-week recovery period, implanted animals began training in voluntary head restraint.

## Psychophysical reverse correlation

The choice-conditioned reverse correlation analysis reveals the degree to which flashes in each time bin contribute to the decision. We selected trials in the final training stage that had the same generative probabilities ($\gamma$) for generating flashes (p(flash on high-probability side) = 0.7; p(flash on low-probability side) = 0.3). We computed the average number of left ($\overline{f}_L(t)$) and right ($\overline{f}_R(t)$) flashes in each time bin ($t$) conditioned on whether the animal went right or left, to obtain the average difference in flash number:

$$\overline{f}_{went\_right}(t) = \overline{f}_R(t) - \overline{f}_L(t) \qquad (1)$$

$$\overline{f}_{went\_left}(t) = \overline{f}_L(t) - \overline{f}_R(t) \tag{2}$$

We next computed the expected difference in flash number for each trial (0.7–.3 = 0.4 for correct trials, -0.4 for error trials), and averaged across trials conditioned on choice. We subtracted this number (the average expected difference in flash number in each bin) from the observed average difference in flash number:

$$\overline{F}_{went\_right}(t) = \overline{f}_{went\_right}(t) - \overline{f}(t)|\gamma \tag{3}$$

$$\overline{F}_{went\_left}(t) = \overline{f}_{went\_left}(t) - \overline{f}(t)|\gamma \tag{4}$$

We multiplied $\overline{F}_{went\_left}$ by -1 so that the two vectors ($\overline{F}$) were both in units of excess right flashes. $\overline{F}_{went\_right}$ and $\overline{F}_{went\_left}$ are the red and blue lines plotted in *Figure 2B*. Positive/negative values in $\overline{F}$ are time bins in which more right/left clicks occurred than expected by chance, respectively.

## 8-parameter sequential sampling model

To obtain a moment-by-moment description of the decision process, we implemented a behavioral model fit to the trial-by-trial data that has been described previously (*Brunton et al., 2013*; *Hanks et al., 2015*). On each trial the model converts the incoming stream of discrete left and right flashes into a scalar quantity *a(t)* that represents the gradually accumulating difference between flashes presented to the two sides. At the end of the trial, the model predicts whether the animal would go right or left if *a* is positive or negative, respectively. The rat's behavior is used to fit parameters that govern how *a(t)* evolves. These parameters quantify sensory noise and noise associated with time, leakiness/instability of the accumulation process, sensory depression/facilitation, side bias, and a lapse rate that corresponds to a fraction of trials on which a random choice is made. The dynamics of *a(t)* are implemented by the following equation:

$$da = \sigma_a dW + (\delta_{t,tR} * \eta_R * C - \delta_{t,tL} * \eta_L * C)dt + \lambda a dt \tag{5}$$

where $\delta_{t,tR,L}$ are delta functions at the flash times, η are Gaussian variables drawn from $N(1, \sigma_s)$, dW is a white-noise Wiener process, and *C* parametrizes adaptation/facilitation of subsequent flashes. Adaptation/facilitation dynamics of *C* are implemented by the following equation:

$$\frac{dC}{dt} = \frac{1-C}{\tau_\phi} + (\phi - 1)C(\delta_{t,tR} + \delta_{t,tL}) \tag{6}$$

In addition, a lapse rate parameter represents the fraction of trials on which the rat responds randomly.

## Signal detection theory model

To describe how noise in the decision process scales with number of flashes, we implemented a model to estimate the width of the distribution of animals' internal estimates of flash number. This signal detection theory-based model assumes that on a given trial, the animal's estimate of flash number presented to each side is a random variable drawn from a Gaussian distribution whose mean is the number of flashes on that side, and whose variance is a free parameter in the model. The difference of those Gaussians predicts the subjects' performance: correct trials occur when the difference of Gaussians is positive (i.e. the random variable drawn from the distribution representing the larger number is in fact larger than the random variable drawn from the distribution representing the smaller number). This was implemented by the following equation:

$$p(correct) = \int_0^\infty N\left(L - S, \sqrt{\sigma_L^2 + \sigma_S^2}\right) d(L - S) \tag{7}$$

Where L and S are the means of the distributions for the larger and smaller numbers, respectively. $\sigma_L$ is the standard deviation of noise when L stimuli has been shown for the 'larger/correct' side, and $\sigma_S$ is the standard deviation of noise when S stimuli has been shown for the 'smaller/incorrect' side. In other words, $\sigma_L^2$ and $\sigma_S^2$ are the variances of the distributions of the numerical estimates of L and S. The values of $\sigma_L^2$ and $\sigma_S^2$ that maximized the likelihood of the animals' behavioral choices were fit

using the Matlab function fmincon. The values of $\sigma_N^2$ reported in this paper were derived using a bootstrapping approach. One thousand surrogate data sets were created by selecting behavioral trials at random with replacement from the original data set. The signal detection theory-based model was then fit to each surrogate data set, producing 1000-fold estimates for $\sigma_0^2 \ldots \sigma_{15}^2$. The $\sigma_n^2$ values reported in the paper represent the mean of over these values and the confidence intervals were derived from the standard deviation of these values.

## Model comparison

To evaluate the goodness-of-fits of the different models, we implemented a nonparametric permutation test. First, for each model, we computed a distribution of bootstrapped $r^2$ values by resampling the behavioral trials with replacement, performing the fitting procedure, and computing the $r^2$ value, on 1000 iterations. For pairwise comparisons, (for example, between the scalar variability (SV) and linear variance (LV) models in *Figure 3F*), the null hypothesis was that the $r^2$ values for each distribution derive from a common distribution. To test this hypothesis, we combined the bootstrapped $r^2$ values from the SV and LV models into a single distribution, and from that combined distribution, created two arbitrary distributions of fake SV and LV $r^2$ values, and computed the average of those arbitrary distributions. We repeated this procedure for many (100, 1000, and 10000) iterations to compute a distribution of arbitrary SV and LV $r^2$ values and consequently, the difference of those arbitrary distributions, which represents the null hypothesis. We treated the area under the null distribution corresponding to the difference between the true SV and LV $r^2$ distributions as the p-value. This procedure yielded identical results with 100, 1000, and 10000 permutations. It was performed for the model comparisons in *Figure 3F*, as well as for evaluation of the SV vs. LV models for the auditory (clicks) data.

To compare the different versions of the signal detection theory-based model (*Figure 4*), for example single accumulator version vs two accumulator version or two accumulator version vs. the two accumulator time-varying version, we used a bootstrapping approach. One hundred surrogate datasets were created by selecting behavioral trials at random with replacement from the original dataset. The signal detection theory-based model was then fit to each surrogate dataset, and the log likelihood for the best-fit parameters was recorded. This gave us a distribution of log likelihoods for each version of the model. We then performed the nonparametric permutation test described above on the distributions of log likelihoods to make pairwise comparisons between the best-fit model and all other models.

## Acknowledgments

The authors would like to thank Klaus Osorio and Jovanna Teran for assistance with behavioral training and animal husbandry. Bingni Brunton contributed data that were collected for a previous paper (*Brunton et al., 2013*). Tim Hanks provided valuable comments on the manuscript and was very helpful in early stages of the analysis. Sam Lewallen and Mikio Aoi suggested useful analytic approaches. In addition the authors would like to thank all members of the Tank and Brody labs for useful discussions.

## Additional information

### Funding

| Funder | Grant reference number | Author |
| --- | --- | --- |
| National Institutes of Health | 5F32NS78913 | Ben B Scott |
| Helen Hay Whitney Foundation | Postdoctoral fellow | Christine M Constantinople |
| National Institutes of Health | R21NS082956 | David W Tank<br>Carlos D Brody |
| National Institutes of Health | U01NS090541 | David W Tank<br>Carlos D Brody |
| Howard Hughes Medical Institute | Investigator | Carlos D Brody |

The funders had no role in study design, data collection and interpretation, or the decision to submit the work for publication.

### Author contributions

BBS, Conception and design, Acquisition of data, Analysis and interpretation of data, Drafting or revising the article; CMC, Acquisition of data, Analysis and interpretation of data, Drafting or revising the article; JCE, Conception and design, Acquisition of data; DWT, Conception and design, Drafting or revising the article; CDB, Conception and design, Analysis and interpretation of data, Drafting or revising the article

### Author ORCIDs

Jeffrey C Erlich, http://orcid.org/0000-0001-9073-7986

### Ethics

Animal experimentation: Animal use procedures were approved by the Princeton University Institutional Animal Care and Use Committee (IACUC) (Protocol #1837 and #1853). These procedures were carried out in accordance with the recommendations in the Guide for the Care and Use of Laboratory Animals of the National Institutes of Health.

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

## Appendix 1

Let

$N_R$ = number of Right pulses,
$N_L$ = number of Left pulses,

and let

$$\sigma_N = k_0 + kN. \tag{1}$$

where $k_0$ and $k$ are constants, and $N$ will represent number of pulses on one of the two sides.

Then define

$$P(\text{choice} = \text{Right}|N_R,\ N_L) = \text{erf}\left(\frac{N_R - N_L}{\sqrt{\sigma_{N_R}^2 + \sigma_{N_L}^2}}\right). \tag{2}$$

where

$$\text{erf}(a) \equiv \frac{1}{\sqrt{2\pi}} \int_{-a}^{\infty} e^{-x^2/2} \mathrm{dx}. \tag{3}$$

Consider the case where $kN_L \gg k_0$ and $N_R = 0$ (the easiest "go Left" trials). Under these conditions $k_0$ is negligible, and we obtain

$$P(\text{choice} = \text{Right}|N_R,\ N_L) \approx \text{erf}\left(\frac{-N_L}{\sqrt{k^2 N_L^2}}\right) \tag{4}$$

$$= \text{erf}\left(\frac{-1}{k}\right). \tag{5}$$

A similar expression holds for the easiest "go Right" trials. **Equation (5) corresponds to the "lapse rate" according to the model:** that is, the probability of making an error even in the easiest trials. Notice (a) that the lapse rate is non-zero even though there is no explicit lapse rate parameter and (b) $k_0$ does not enter into the expression for the lapse rate.

Now consider the case where the number of pulses is very low (e.g., $N_R = 0$ and $N_L = 1$). The larger that $k_0$ is, the closer that $P(\text{choice} = \text{Right}|N_R, N_L) \approx 0.5$, in other words, the closer that performance will approach chance. **Consequently, for large $k_0$, performance starts near chance for very short duration trials** (since these will have low numbers of clicks). Performance will grow for longer duration trials, and will reach a maximum fraction correct of $1 - \text{erf}(-1/k)$ for the easiest, longest, trials.

