## [Decision Letter]

Thank you for submitting your work entitled "Sources of noise during accumulation of evidence in unrestrained and voluntarily head-restrained rats" for peer review at *eLife*. Your submission has been favorably evaluated by Eve Marder (Senior Editor), Maoshige Uchida (Reviewing Editor), and three reviewers, one of whom, Patrick Simen, has agreed to share his identity.

The reviewers have discussed the reviews with one another and the Reviewing Editor has drafted this decision to help you prepare a revised submission.

All the reviewers thought that the experiments are well done, the analyses are solid (but see below), and the conclusions are very interesting. In particular, the reviewers thought that the use of a decision paradigm in which the exact information provided to an animal on each trial can be precisely controlled is a powerful means to study the mechanisms underlying decision making. This property indeed allowed the authors to analyze the decision-making process in a greater detail. The authors conclude that the standard deviation (SD) of the internal estimates grows linearly with the number of flashes (i.e. features Weber scaling), which violates the linear scaling in variance (rather than SD) commonly assumed by evidence accumulation models (including the one used in Brunton et al. [2013]). On the other hand, the reviewers raised some important issues that the authors should address before publication in *eLife*.

Essential revisions:

1) Model incompatibility. Some of the important conclusions are based on the fit parameters obtained using a drift diffusion (DDF) model (Brunton et al., 2013). However, the authors' analysis using the signal detection theory shows that some of the important assumptions in DDF models are wrong. How can the authors justify the conclusions obtained by the DDF model? The authors should analyze the data further to address the issue of whether and how the incompatibility between these models affects the validity of their conclusions.

2) The authors should perform model comparisons more quantitatively e.g. using AIC/BIC. The reviewers raised the issue that unlike what is stated in the text, the models might have different numbers of parameters. Please clarify. Also, model comparisons should be done at the level of individual animals as well as populations.

3) The reviewers thought that some of the assumptions such as zero drift during inter-pulse intervals are not justified.

4) The authors should show more results regarding the variability between animals.

5) How bootstrap methods were applied needs to be explained more explicitly. If the analyses were run such that the number of resampling affects the significance level, then the statistical results (p-value) do not appear to be valid. Please clarify.

6) The model, as well as the data, seems to suggest that scaling of the numbers of pulses by the same factor will lead to identical discriminability. This property seems important. Please clarify whether this is true in the model as well as the data.

---

## [Author Response]

*1) Model incompatibility. Some of the important conclusions are based on the fit parameters obtained using a drift diffusion (DDF) model (Brunton et al., 2013). However, the authors' analysis using the signal detection theory shows that some of the important assumptions in DDF models are wrong. How can the authors justify the conclusions obtained by the DDF model? The authors should analyze the data further to address the issue of whether and how the incompatibility between these models affects the validity of their conclusions.*

We thank the reviewers for helping us greatly clarify this aspect of the original manuscript. We have revised the current manuscript to address this issue using the following approaches:

A) The critical features used in our signal detection theory (SDT) model are that diffusion noise in the inter-pulse interval is near-zero, and that pulses from throughout the entire stimulus interval contribute roughly equally to the animal’s decision. We have performed additional model-free approaches to specifically support these claims (Figure 2; Figure 2—figure supplement 2, Figure 2—figure supplement 3) without reference to the Brunton et al. model, and we have rewritten the text to make it clear that our conclusions are based primarily on model-free analyses, and do not require the DDF model.

B) We have rewritten the text to place less emphasis on the DDF model from Brunton et al., and as above, to make it clear that the SDT model and its conclusions do not depend on the Brunton et al. model. As the reviewers point out, one of the main messages of our paper is that the Brunton et al. model made some unjustified assumptions, so it makes no sense to depend on it.

We nevertheless still include, in the supplementary material, the results of applying the Brunton et al. model to our visual task data, so as to facilitate direct comparison between analyses of our task and the related previous auditory pulses task of Brunton et al. (It is interesting to note that the Brunton et al. fits for our visual task are qualitatively consistent with the fits to the previous auditory task – i.e., that model alone is not sufficient to distinguish qualitative differences between the two tasks.) Similarly, we report the results of applying our SDT model to both the visual task data and the auditory task data.

It is also interesting to note that, although imperfect, the interpretations that can be drawn from the Brunton et al. DDF model are consistent with the model-free analyses on which the paper now rests. We hope that readers interested in drift diffusion-like models of decision making will find the aspects of the DDF model that are robust across studies – as well as its shortcomings – informative.

*2) The authors should perform model comparisons more quantitatively e.g. using AIC/BIC. The reviewers raised the issue that unlike what is stated in the text, the models might have different numbers of parameters. Please clarify. Also, model comparisons should be done at the level of individual animals as well as populations.*

We thank the reviewers for pointing this out: they are quite right, in our original submission we did indeed compare models that had different numbers of parameters but failed to use metrics that penalize model complexity, such as AIC. The question of which specific penalization to use for comparing non-nested models with different parameters (AIC, BIC, DIC, “empirical bayes” etc) is a subject of ongoing debate within the machine learning literature. The reviewers’ comment prompted us to perform a new analysis, which we feel is far simpler and does not suffer from the complications of statistical model comparison for non-nested models. We simplified our models so that all of the 8 model variants that we compare have exactly the same number of parameters (only 2), with similar structures for the two parameters across all the 8 models. This approach made comparison across models much more straightforward, and allowed us to compare log likelihoods directly. The results of this new comparison are described in our Results section and in our revised Figure 4. We also include a new figure (Figure 4—figure supplement 1) that compares the different models for each individual animal.

*3) The reviewers thought that some of the assumptions such as zero drift during inter-pulse intervals are not justified.*

As we understood it, the reviewers’ suggestion here is that a flash might create a long-lasting input to the accumulator (i.e., non-zero drift during inter-pulse intervals), whereas we assumed zero drift during inter-pulse intervals. It is worth noting that if a pulse created a short-lasting non- zero drift (e.g., on the order of tens of milliseconds), our results would remain essentially the same, because all analyses and results depend on the integrated effect of each pulse. The important question is thus with respect to long-lasting non-zero drift.

The longer a non-zero post-pulse drift, the greater its cumulative impact on the accumulator. Thus we would expect decisions to be biased toward the side with the greatest cumulative post-pulse time. To examine whether our assumption of zero drift during inter-pulse intervals was reasonable, we implemented a regression model with two parameters: cumulative post-pulse interval duration and difference in flash number (plus a third parameters, for a constant term). Long-lasting post-pulse drift would produce a significant positive regression weight for the cumulative post-pulse duration. Instead, the model revealed that the rats’ choices were better explained by differences in flash number on each trial. More specifically, the regression coefficient for the difference in flash number was. 25 +/- .0038, compared to -.06 +/- .0036 for cumulative post-pulse interval duration (and. 02 +/- .0012 for the bias/constant term). We concluded that nonzero drift during the inter-pulse intervals did not appreciably contribute to the rats’ behavior.

*4) The authors should show more results regarding the variability between animals.*

We thank the authors for this suggestion. We have included twelve additional supplementary figures to address this issue. Figure 1—figure supplement 2 illustrates the psychometric performance of each individual rat. Figure 2—figure supplement 1 plots the best-fit model parameters to the drift diffusion-like model from Brunton et al. (2013) for each rat. Figure 2—figure supplement 2 and Figure 2—figure supplement 3 illustrate the effects of flash number and trial duration on behavioral performance for each individual rat. Figure 2—figure supplement 4 illustrates the psychophysical kernel (i.e. reverse correlation) of each individual rat. Figure 3—figure supplement 1 plots the best-fit model parameters to the signal-detection theory model for each rat. Figure 3—figure supplement 2 compares the signal- detection theory model prediction to the data from each rat. We have also included additional analyses of each rat that performed the auditory (clicks) version of this task. Figure 3—figure supplement 4 plots the fits of the SDT model for each rat from the clicks task, and Figure 3—figure supplement 5 plots the result of a permutation test for each clicks rat comparing the goodness-of-fit of scalar variability versus linear variance. Figure 3—figure supplement 6 and Figure 3—figure supplement 7 plot the chronometric curve of each rat, as well as the signal-detection theory model prediction. Figure 4—figure supplement 1 compares model performance of all eight models described in Figure 4 for each rat in the visual version of the task. We have referenced these analyses in the main text and included appropriate supplementary figure legends.

*5) How bootstrap methods were applied needs to be explained more explicitly. If the analyses were run such that the number of resampling affects the significance level, then the statistical results (p-value) do not appear to be valid. Please clarify.*

We thank the reviewers for pointing this out. We have recomputed significance using a nonparametric permutation test, so that the p-value is not affected by amount of resampling, which we confirmed by observing that the procedure yielded the same results when performed with 100, or 1000, or 10,000 permutations.

In more detail, when comparing the distribution of goodness of fit values (R2) for the scalar variability (SV) and linear variance (LV) models in Figure 3, the null hypothesis is that the R2 values for each distribution derive from a common distribution. [To clarify: the distribution of bootstrapped R2 values were obtained by resampling from the behavioral trials with replacement, performing the fitting procedure, and computing the R2 value, on 1000 iterations.] To test the null hypothesis, we combined the bootstrapped R2 values from the SV and LV models into a single distribution, and from that combined distribution, created two arbitrary distributions of fake SV and LV R2 values, and computed the average of those arbitrary distributions. We repeated this procedure for many iterations to compute a distribution of arbitrary SV and LV R2 values and consequently, the difference of those arbitrary distributions, which represents the null hypothesis. The area of the null distribution corresponding to the difference between the true SV and LV R2 distributions is the p-value. As pointed out above, the procedure yielded identical results with 100, 1000, and 10000 permutations. It was performed for the model comparisons in Figure 3, as well as for evaluation of the SV vs. LV models for the auditory (clicks) data. Additionally, to evaluate the models in Figure 4, this permutation test was performed on the distributions of bootstrapped likelihoods for each model. There is a new section of the Methods section that describes the bootstrapping and permutation procedures in full (see Methods: Model comparison).

*6) The model, as well as the data, seems to suggest that scaling of the numbers of pulses by the same factor will lead to identical discriminability. This property seems important. Please clarify whether this is true in the model as well as the data.*

We thank the reviewers for suggesting this very interesting analysis: they are indeed correct that it is an important property, and it led us to some further analyses that, among other things, helped to better understand the comparison between our visual task and the auditory task of Brunton et al.

First, we have included a new supplementary figure (Figure 3—figure supplement 3) confirming that, as predicted by the reviewers, scaling the number of pulses on both sides by the same factor leads to nearly identical performance. This is true in both the data and in the signal-detection theory model predictions. (The signal detection theory model does predict some deviations from scalar variability, at very low numbers of flashes. These are also observed in the data, as predicted, and are also now described in the main text).

Second, we observed that perfect scalar variability would imply that for the auditory task of Brunton et al., performance should not increase for longer stimulus durations; yet in the data itself, performance does clearly increase. When fitting the auditory data with the SDT model, we found that a linear relationship between noise standard deviation and number of pulses still applied (as in scalar variability), but with a significant non-zero offset k0, i.e., σN = k0 + kN. This non-zero k0 can account for the observed improvement in performance over time in the auditory data, and contrasts with data from the visual task, for which k0 ≈ 0 (compare Figure 3—figure supplement 6 with Figure 3—figure supplement 7).

Both of these are important further analyses that followed from the reviewers’ observations and that we think readers will find interesting. In particular, these analyses allowed us to strengthen comparisons between the current visual data and the previous auditory data, and they establish that the linear relationship σN = k0 + kN is not specific to the visual sensory modality.

The new main text discussing these points can be found in the section Results: “Subjects’ estimates of flash number exhibit scalar variability”.